# On the Uniqueness of the Standard Genetic Code

**DOI:** 10.3390/life7010007

**Published:** 2017-02-13

**Authors:** Gabriel S. Zamudio, Marco V. José

**Affiliations:** Theoretical Biology Group, Instituto de Investigaciones Biomédicas, Universidad Nacional Autónoma de México, México D.F. 04510, Mexico; gazaso92@gmail.com

**Keywords:** RNY code, Standard genetic code, evolution of the genetic code, frozen code, degeneracy, aminoacyl-tRNA synthetases, symmetry

## Abstract

In this work, we determine the biological and mathematical properties that are sufficient and necessary to uniquely determine both the primeval RNY (purine-any base-pyrimidine) code and the standard genetic code (SGC). These properties are: the evolution of the SGC from the RNY code; the degeneracy of both codes, and the non-degeneracy of the assignments of aminoacyl-tRNA synthetases (aaRSs) to amino acids; the wobbling property; the consideration that glycine was the first amino acid; the topological and symmetrical properties of both codes.

## 1. Introduction

A fundamental feature of all life forms existing on Earth is that, with several minor exceptions, they share the same standard genetic code (SGC). This universality led Francis Crick to propose the frozen accident hypothesis [1], i.e., the SGC does not change. According to Crick [1], the SGC code remained universal because any change would be lethal, or would have been very strongly selected against and extinguished.

The astonishing diversity of living beings in the history of the biosphere has not been halted by a frozen SGC. The inherent structure of the frozen SGC, in concert with environmental influences, has unleashed life from determinism.

It is widely accepted that there was an age in the origin of life in which RNA played the role of both genetic material and the main agent of catalytic activity [1,2,3]. This period is known as the RNA World [4,5].

The reign of the RNA World on Earth probably began no more than about 4.2 billion years ago, and ended no less than about 3.6 billion years ago [6]. Eigen and coworkers (1968) [7] revealed kinship relations by alignments of tRNA sequences and they concluded that the genetic code is not older than but almost as old as our planet. There is an enormous leap from the RNA World to the complexity of DNA replication, protein manufacture and biochemical pathways. Code stability since its formation on the early Earth has contributed to preserving evidence of the transition from an RNA World to a protein-dependent world.

The transfer RNA (tRNA) is perhaps the most important molecule in the origin and evolution of the genetic code. Just two years after the discovery of the double-helix structure of DNA, Crick [8,9] proposed the existence of small adaptor RNA molecules that would act as decoders carrying their own amino acids and interacting with the messenger RNA (mRNA) template in a position for polymerization to take place. 

The SGC is written in an alphabet of four letters (C, A, U, G), grouped into words three letters long, called triplets or codons. Crick represented the genetic code in a two-dimensional table arranged in such a way that it is possible to readily find any amino acid from the three letters, written in the 5′ to 3′ direction of the codon [1]. Each of the 64 codons specifies one of the 20 amino acids or else serves as a punctuation mark signaling the end of a message. 

Crick proposed the wobble hypothesis [10,11], which accounts for the degeneracy of the SGC: the third position in each codon is said to wobble because it is much less specific than the first and second positions.

Given 64 codons and 20 amino acids plus a punctuation mark, there are $21^{64}\approx 4\times 10^{84}$ possible genetic codes. This staggering number is beyond any imaginable astronomical number, the total count of electrons in the universe being well below this number. Note, however, that this calculation tacitly ignores the evolution of the SGC. If we assume two sets of 32 complementary triplets where each set codes for 10 amino acids, we would have $10^{32}\times 10^{32}=10^{64}$ possible codes. Then we have a reduction of the order of $4\times 10^{20}.$ Albeit this is a significant reduction, it is still a very large number. Many more biological constraints are necessary. The result that only one in every million random alternative codes is more efficient than the SGC [12] implies that there could be $∼4\times 10^{78}$ genetic codes as efficient as the SGC. This calculation does not offer deeper insights concerning the origin and structure of the SGC, particularly the frozen accident.

Crick [1] argued that the SGC need not be special at all; it could be nothing more than a “frozen accident”. This concept is not far away from the idea that there was an age of miracles. However, as we show in this article, there are indeed several features that are special about the SGC: first, it can be partitioned into two classes of aminoacyl-tRNA synthetases (aaRSs) [13]; secondly, the SGC can be broken down into a product of simpler groups reflecting the pattern of degeneracy observed [14,15]; third, it has symmetrical properties, and evolution did not erase its own evolutionary footsteps [16].

Several models on the origin of the genetic code from prebiotic constituents have been proposed [17,18,19,20,21]. Among the 20 canonical amino acids of the biological coding system, the amino acid glycine is one of the most abundant in prebiotic experiments that simulate the conditions of the primitive planet, either by electrical discharges or simulations of volcanic activity [22,23,24], and this amino acid is also abundant in the analysis of meteorites [25]. Bernhardt and Patrick (2014), and Tamura (2015) [26,27] also suggested that glycine was the first amino acid incorporated into the genetic code according to an internal analysis of its corresponding tRNA and its crucial importance in the structure and function of proteins. Part of this abundance can be ascribed to its structural simplicity when compared with the structure of the remaining 19 canonical amino acids. Several models for the origin of the coding system mirror glycine as one of the initial amino acids in this system [26,27,28,29]. 

The SGC was theoretically derived from a primeval RNY (R means purine, Y pyrimidine, and N any of them) genetic code under a model of sequential symmetry breakings [14,15], and vestiges of this primeval RNY genetic code were found in current genomes of both Eubacteria and Archaea [16]. All distance series of codons showed critical-scale invariance not only in RNY sequences (all ORFs (Open reading frames) concatenated after discarding the non-RNY triplets), but also in all codons of two intermediate steps of the genetic code and in all kind of codons in the current genomes [16]. Such scale invariance has been preserved for at least 3.5 billion years, beginning with an RNY genetic code to the SGC throughout two evolutionary pathways. These two likely evolutionary paths of the genetic code were also analyzed algebraically and can be clearly visualized in three, four and six dimensions [15,30,31].

The RNY subcode is widely considered as the primeval genetic code [32]. It comprises 16 triplets and eight amino acids, where each amino acid is encoded by two codons. The abiotic support of the RNY primeval code is in agreement with observations on abundant amino acids in Miller’s sets [33] and in the chronology of the appearance of amino acids according to Trifonov’s review [34]. It has been shown that once the primeval genetic code reached the RNY code, the elimination of any amino acid at this stage would be strongly selected against and therefore the genetic code was already frozen [35].

There are 20 aaRSs which are divided into two 10-member, non-overlapping classes, I and II, and they provide virtually errorless aminoacylation of tRNAs [36,37]. Therefore, this operational code is non- degenerate [36,37]. 

In this work, we pose the following question: What are the minimum necessary and sufficient biological and mathematical properties to uniquely determine the primeval RNY code and the SGC?

## 2. Mathematical Model of the RNY Code

The RNY code consists of codons where the first base is a purine (R), the third is a pyrimidine (Y) and the second is any of them (Table 1). In this code, the wobble position is strictly present on the third base of the triplet. The number of possible RNY codes is $8^{16}=2.81\times 10^{14}.$

The SGC has been represented in a six-dimensional hypercube [30,38]. Observing that 64 is equal not only to 4^3^ but also to 2^6^, the codon table can be organized as a six-dimensional hypercube [30]. In such a model, the set of codons are treated as the 64 vertices of the hypercube, and they are joined by edges which connect codons that differ by a single nucleotide. Each dimension describes a type of mutation, transition or transversion acting on each of three bases of any codon. Consequently, we obtain the six dimensions. 

This symmetrical model [38] can be partitioned *exactly* into two classes of aaRSs in six dimensions; it displays symmetry groups when the polar requirement is used, and the SGC can be broken down into a product of simpler groups reflecting the pattern of degeneracy observed, and the salient fact that evolution did not erase its own evolutionary footsteps. The symmetrical model and the Rodin-Ohno model [13] are one and the same [38].

Similarly, the RNY subcode can be represented in a four-dimensional hypercube (Figure 1). This hypercube will be employed to reduce the possible number of mappings, by considering its topology and neighborhood properties. Codons that codify the same amino acid are neighbors. Note from Figure 1 that codons for the same amino acid are next to each other, due to the fact that they differ in only the third base and therefore they are at distance of one. A detailed description of the 6D hypercube representing the SGC can be found in Reference [30]. 

## 3. Combinatorics of the RNY Code

We have noted above that the number of possible codes composed by eight amino acids and 16 triplets is $8^{16}=2.81\times 10^{14}.$ This number includes codes completely redundant (all codons assigned to the same amino acid) or codes in which all amino acids share the same degeneration, as in the present RNY code. Also, there may not be restrictions between the two classes of aaRS and their corresponding amino acids. First, we consider the restriction in which all amino acids are coded by two triplets, and such codes are given by the multinomial coefficient $(2,2,2,2,2,2,2,2)!=\frac{16!}{2{!}^{8}}≃8.17\times 10^{10}.$ The present RNY code arranges the triplets so that two codons for the same amino acid are neighbors in the four-dimensional cube. With such a restriction, there are $\left(\begin{array}{l} 4\\ 1 \end{array}\right)8!=161,280$ possible RNY codes, since there are four possible configurations in which amino acids can be arranged in the 4-dimensional hypercube hypercube. This neighborhood property preserves the degeneracy irrrespective of the particular wobbling nucleotide, not necessarily the third position. The number 8! accounts for the fact that all the permutations in the assignation of amino acids maintain the property that the two codons that encode the same amino acid must be neighbors.

Considering the third base as the source of variability in the code, the number of posibilities is reduced to 8!=40,320. If we consider only the first two bases that determine the amino acid, it is possible to reduce the four-dimensional cube to a three-dimensional cube in which the vertices represent the first two nucleotides (Figure 2a). If the vertices are relabeled to show the codified amino acid, we obtain a phenotypic cube (Figure 2b). 

If we consider that there are two amino acids that belong to class I and six amino acids that correspond to class II, then there are $2\left(\begin{array}{l} 8\\ 2 \end{array}\right)6!=37,440$ possible codes. This calculation comes from taking two out of the eight amino acids and assigning them to class I, considering its permutations and also the permutations of the amino acids of class II. To maintain the topological properties of the RNY model, four triplets of class I must form a square in the four-dimensional model, or similarly, the dinucleotides must be neighbors, i.e., they are connected by an edge in the cube representation. In this case, there are 2(12)6!=17,280 different codes that preserve the aaRSs distribution in the code and in the model. This number arises from the 12 edges available in the cube to join classs I amino acids and the permutations of classs II amino acids.

In order to maintain the topological properties of the three-dimensional cube, the amino acids of a code must share the neighboring properties of the current RNY code. In other words, if two amino acids are next to each other in the current model, then they are also adjacent in a model constructed by such a code. This property is manifested by the fact that such codes are built by the symmetries of the present model, so that there are 48 different codes that keep the topology of the current code intact.

The ocurrence of glycine as the first amino acid and its assignment to the triplets GGC and GGU as a fixed starting point in the evolution of the SGC impose another restriction, particularly when contrasted with the topology of the four- and three-dimensional cubes, since it fixates isoleucine to AUC and AUU in order to keep the adjacency properties. In this case, there are as many as $\left(\begin{array}{l} 3\\ 1 \end{array}\right)2=6$ possible codes, due to the fact that there are three possible positions for valine that maintain its adjacency to isoleucine, and there are two symmetrical configurations (given by a reflection) that maintain the rest of the topology.

In the actual code, all triplets where the middle base is uracile codify for amino acids of class I, and this pattern forces the triplets of valine to be GUC and GUU, which in turn also fixes AGC and AGU for serine. This results in two possible RNY codes, which here and further on will be denoted by ○RNY and ∅RNY. The ○RNY denotes the actual and original RNY code, whereas ∅RNY represents an alternative code in which the codons for threonine and alanine are simultaneously interchanged with the ones of aspartic acid and asparagine, respectively. The fixation of another amino acid would completely constraint the number of RNY possible codes to only one!

## 4. Evolution of the RNY Code by Means of Frame-Shifts and Transversions

Two genetic codes from which the primeval RNA code could have originated the SGC were derived [14,15,16]. The primeval RNA code consists of 16 codons that specify eight amino acids (then this code shows a slight degeneration). The extended RNA code type I consists of all codons of the RNY type plus codons obtained by considering the RNA code, but in the second (NYR-type) and third (YRN-type) reading frames. The extended RNA code type II comprises all codons of the RNY type plus codons that arise from transversions of the RNA code in the first (YNY type) and third (RNR) nucleotide bases. Then, by allowing frame-reading mistranslations, we arrived at 48 codons that specify 17 amino acids and the three stop codons. If transversions in the first or third nucleotide bases of the RNY pattern are permitted, then there are also 48 codons that encode for 18 amino acids but no stop codons.

In the context of the frozen concept, it was concluded that considering the symmetries of both extended RNA codes, the primeval RNY code was already frozen and it evolved like a replicating and growing icicle [14]. The composition of both extended codes eventually leads to the actual SGC.

As the RNY is described mathematically as a four-dimensional cube, each extended code comprises a duplication of the RNY cube in order to determine a five-dimensional prism as an intermediate step towards the final six-dimensional cube for the SGC. Supposing one of the two alternative RNY codes as the initial code, the number of possible extended codes can be calculated. Then, assuming, as before, that wobbling occurs principally at the third base, the current degeneration of the code and the topology given by the mathematical model shall be maintained.

If the ○RNY is used as a cornerstone for the formation of the genetic code, then, regardless of the evolutionary path chosen, there are two SGCs which are compatible with all the assumptions. These are the actual SGC and a second one in which the codifications of AUG and UGG are interchanged with the ones of AUA and UGA, respectively. These modifications make it so that methionine is codified by AUA and tryptophane by UGA, while AUG codes for isoleucine and UGG is a stop signal. The rest of the code remains unaltered.

On the other hand, if ∅RNY is used as an initial condition, then there are no possible codes on any evolutionary path which meet all hypotheses. In other words, it is not possible to derive the SGC from ∅RNYwithout violating at least one of the considered properties. This is due to the fact that the mathematical model forbids the possible extended codes that would keep biological properties such as wobbling and the binary division of aaRSs.

## 5. Discussion

It is possible to gradually add properties to the RNY code to reduce the number of possible codes from 2.81 × 10^4^ to only one. This is done when considering the current properties of degeneracy of the RNY code and the wobble, the aaRSs distribution in the RNY and in the SGC, and finally the mathematical model to represent the genetic code and its induced property of adjacency. The mathematical model plays an important role in the reduction of the possible number of codes. The 37,440 possible RNY codes were obtained by considering the degeneration in the third base and by assuming that the distribution of aaRRs classes is the same as in the current RNY code. Further reductions, up to one code, were only accomplished by the use of our mathematical model. Both evolutionary paths majorly reduce the number of possible genetic codes from the staggering number of $4.18\times 10^{84}$ to only two, which consists of the current code and an alternative code with a subtle modification. The alternative RNY code, ∅RNY, cannot lead to an SGC that is compatible with all the hypotheses by means of the transversions and frame-shift reading mistranslations. Hence, the SGC evolved from the ○RNY code.

Novozhilov et al. [39] found that the SGC is a suboptimal random code in regard to robustness to error of translations. Thus, the SGC appears to be a point on an evolutionary trajectory from a random code about halfway to the summit (or to the valley) of the local peak in a rugged fitness landscape. 

So far, all we know is terrestrial biology. If life is to be found somewhere else in the universe, and even if its ancestry can be traced back to primitive organisms, the rules of the assignments of codons to amino acids may not necessarily be the same and the amino acids may be even chemically different to those found in known terrestrial life. Different environments and different evolutionary paths on different worlds could result in completely different genetic codes and patterns of evolution.

In conclusion, the SGC is certainly ubiquitous in Earth, and what we would expect to find in living beings on other planets is, precisely, this universal biological property: a genetic coding system.

## Figures and Tables

**Figure 1 life-07-00007-f001:**
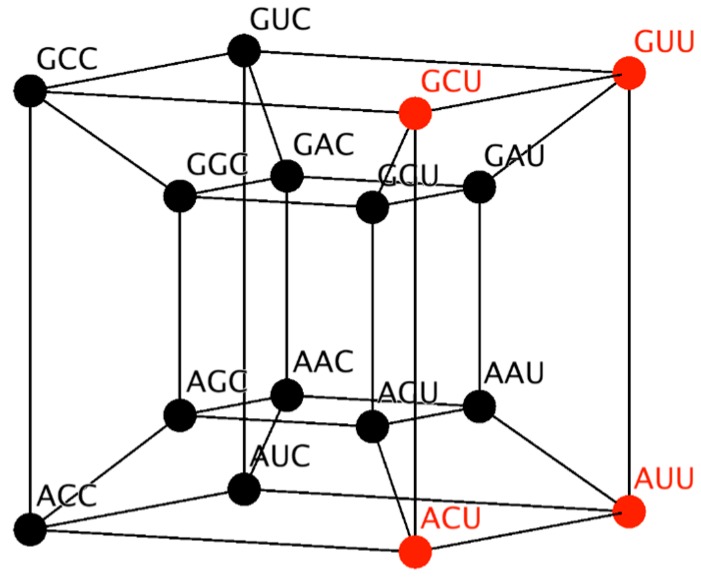
Four-dimensional hypercube that represents the RNY code. Codons for amino acids of class I are in red and those for class II are in black.

**Figure 2 life-07-00007-f002:**
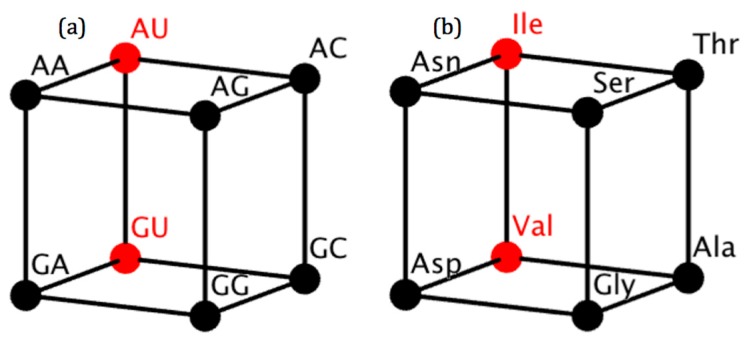
(**a**) Cube of RNY dinucleotides according to the four-dimensional model of the code. Dinucleotides for class I amino acids are in red; and those for class II are in black; (**b**) Phenotypic cube of amino acids according to the four-dimensional model of the RNY code. Class I amino acids are in red and those of class II are in black.

**Table 1 life-07-00007-t001:** RNY code. Amino acids that pertain to class I are in red, and those that correspond to class II are in black.

Amino Acid	Codons	Amino Acid	Codons
Asn	AAC, AAU	Thr	ACC, ACU
Asp	GAC, GAU	Ala	GCC, GCU
Ser	AGC, AGU	Ile	AUC, AUU
Gly	GGC, GGU	Val	GUC, GUU

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
