# Peer review of "On the Uniqueness of the Standard Genetic Code"

_life, 2017, doi:10.3390/life7010007_

Round 1
Reviewer 1 Report
The authors present mathematical arguments about the properties of RNY genetic (sub)codes. In particular they aim to provide arguments that, by adding contraints on the combinatorics of codon - amino acid assignment, only a small amount of assigments (i.e. codes) remain, supporting the current standard genetic code.
In general the manuscript seems to be suitable for publication, there are some assuptions included that needs to be made clear for the reader, and also relevant literature needs to be added.
1. There exists a number of theories in theoretical biology about the evolution of the genetic code. Cricks "frozen accident" is just one beside e.g. "stereochemical affinity" and others.
Early fixation might be plausible (cf. Freeland SJ, Knight RD, Landweber LF, Hurst LD (Apr 2000). "Early fixation of an optimal genetic code". Molecular Biology and Evolution. 17 (4): 511–18. ) but other theories are also possible.
The manuscript would greatly benefit from discussing the relation of the results obtained and how they fit into the already proposed theories.
2. A potential "problem"/"fallacy" might lie in the fact that the authors used the standard genetic code (as observed today) and identified the 16 RNY codons and their actual amino acids. But we don't know whether these assignments were fixed along the whole way along evolutionary paths, or changed in between. The wide distribution (in all actual organisms we know) might support this assumption, but we are not sure how flexible the code might have been while the time of the first common ancestor.
3. on page 2 (line 54) the calculations of all combination of 21 elements (amino acids and stop codon) on 64 codons (=21^64) is used to illustrate the huge amount of potential codes. Of course this calculation is not reasonable, as the authors state. In particular it does not only ignore evolutionary concepts, but also contains a huge number of non-sense codes (e.g. all codons lead to stop codons). The illustrative nature of the calculation should also be made clearer to allow the reader to plausible judge this information.
4. The construction of hypercubes should be expained in a little more detail (one or two more sentences) to allow readers unfamiliar with the notation to follow the manuscript.
5. In the discussion (p6, line 196) the authors state that "it is possible to gradually add properties" to reduce the number of possible codes. This is certainly true in the mathematical model. What is missing here is a discussion about biological/evolutionary mechanisms (and perhaps a listing of references showing this in real experiments) that might relate to these steps within the cell.
6. The last part of the discussion, aiming to illustrate the arbitrariness of the codes assignments using extraterrestrial evolution (life on other planets) is a rather placative and speculative "forecast". There may exists less overstate ways to describe the actual property to authors wanted to illustrate.
7. The authors lack to set their work into a wider relation to other existing work in the field. The references contain many self cites (6 of 19 ) and citations to Crick (5 of 19) . At least in the introduction an overview on already published related literature should be given.
Eg.
- The currentyl published special issue "DNA as Information" http://rsta.royalsocietypublishing.org/content/374/2063
- The works by Tlusty
doi:10.1103/PhysRevLett.100.048101
doi:10.1016/j.plrev.2010.06.002
or
- Rumer (see special issue)
or
- Koonin et al. https://www.ncbi.nlm.nih.gov/pmc/articles/PMC3293468/
and (for all papers) references therein.
The here provided list is only a suggestion of possible references and the authors might identify others better fitting to the RNY model.
Minor comments
p2,line 42: the abbreviation "tRNA" was used before, please explain abbreviation at the first occurence
p5, line 147: The statement "Glycine was the first amino acid" needs a reference.
Author Response
January 18th, 2017
Ms. Emma Li
Assistant Editor
Manuscript ID: life-170440
Title: On the Uniqueness of the Standard Genetic Code
Special Issue: The Origin and Evolution of the Genetic Code: 100th Anniversary Year of the Birth of Francis Crick
Dear Ms. Emma Li,
Thank you for your email dated January 12th, 2017 and for the referee’s comments.
We thank the referees for their patience and careful reading of our paper and for helping us to improve its quality and presentation.
Please find enclosed with this letter the revised version of the manuscript.
We have carefully read the reviewers’ comments of our previous version of the manuscript, and we have rewritten the paper with their suggestions in mind. Reviewer 1 mentions that the article is suitable for its publication. Reviewer 2 accepts the manuscript. Reviewer 2 and Reviewer 3 suggest that more references are needed. Modifications in the manuscript are highlighted in red color, and the answers in this reply letter are marked in red.
We give below the specifics of our response to the reviewers and answer their comments in the same order as their reports.
Reviewer 1
The authors present mathematical arguments about the properties of RNY genetic (sub)codes. In particular, they aim to provide arguments that, by adding constraints on the combinatorics of codon - amino acid assignments, only a small amount of assignments (i.e. codes) remain, supporting the current standard genetic code.
In general, the manuscript seems to be suitable for publication, there are some assumptions included that needs to be made clear for the reader, and also relevant literature needs to be added.
1. There exists a number of theories in theoretical biology about the evolution of the genetic code. Cricks "frozen accident" is just one beside e.g. "stereochemical affinity" and others.
Early fixation might be plausible (cf. Freeland SJ, Knight RD, Landweber LF, Hurst LD (Apr 2000). "Early fixation of an optimal genetic code". Molecular Biology and Evolution. 17(4): 511–18), but other theories are also possible.
The manuscript would greatly benefit from discussing the relation of the results obtained and how they fit into the already proposed theories.
References 2-6, 12, 17-29, and 39 were added to the manuscript. In particular, references 17-21 deal with theories of the evolution of the genetic code.
2. A potential "problem"/"fallacy" might lie in the fact that the authors used the standard genetic code (as observed today) and identified the 16 RNY codons and their actual amino acids. But we don't know whether these assignments were fixed along the whole way along evolutionary paths, or changed in between. The wide distribution (in all actual organisms we know) might support this assumption, but we are not sure how flexible the code might have been while the time of the first common ancestor.
References 1, 7, 13-16, 31-32, and 35 support the RNY subcode as a primeval code.
3. on page 2 (line 54) the calculations of all combination of 21 elements (amino acids and stop codon) on 64 codons (=21^64) is used to illustrate the huge amount of potential codes. Of course, this calculation is not reasonable, as the authors state. In particular, it does not only ignore evolutionary concepts, but also contains a huge number of non-sense codes (e.g. all codons lead to stop codons). The illustrative nature of the calculation should also be made clearer to allow the reader to plausible judge this information.
We added the following in Introduction: “If we assume 2 sets of 32 complementary triplets where each set codes for 10 amino acids, we would have possible codes. Then we have a reduction of the order of Albeit this is a significant reduction it is still a very large number. Many more biological constraints are necessary.
A remark to the reviewer:
Further reduction on the number of possible genetic code according to its degeneracy
The genetic code is, in essence, a surjective function. A function is surjective if, for each for at least one After some calculations we have found that the number of surjective functions is: This is the whole number of possible coding functions, under the assumption that each amino acid, or the stop signal, is only coded by 1, 2, 3, 4 or 6 triplets. This number can still be further reduced in several ways. Given the RNY code, there are 16 triplets and 8 di-codonic amino acids. Hence there are only surjective functions on the set RNY onto the set of the 8 primary amino acids. We are assuming that the 16 codons are equally equipartitioned into binary sets, where the 2 members differ only in the 3rd nucleobase which is either C or U. Hence the probability of appearance of the RNY code, although transient, was very high.
4. The construction of hypercubes should be explained in a little more detail (one or two more sentences) to allow readers unfamiliar with the notation to follow the manuscript.
After Table 1 we added the following sentence:
“Each dimension describes a type of mutation, transition or transversion acting on each of three bases of any codon. Consequently we obtain the six dimensions.”
5. In the discussion (p6, line 196) the authors state that "it is possible to gradually add properties" to reduce the number of possible codes. This is certainly true in the mathematical model. What is missing here is a discussion about biological/evolutionary mechanisms (and perhaps a listing of references showing this in real experiments) that might relate to these steps within the cell.
The steps of our calculation consider the evolution of the SGC from the RNY subcode. We stated in lines 88-90 that: “The SGC has been theoretically derived from a primeval RNY (R means purine, Y pyrimidine, and N any of them) genetic code under a model of sequential symmetry breakings [14-15], and vestiges of this primeval RNY genetic code were found in current genomes of both Eubacteria and Archaea [16].”
6. The last part of the discussion, aiming to illustrate the arbitrariness of the codes assignments using extraterrestrial evolution (life on other planets) is a rather placative and speculative "forecast". There may exists less overstate ways to describe the actual property to authors wanted to illustrate.
Our conclusion is not a “forecast”. We just think that we need to be open minded to very different genetic codes to the ones found in Earth. If life forms are to be found in any other place in the Universe they should possess a genetic code subject to an evolutionary process.
7. The authors lack to set their work into a wider relation to other existing work in the field. The references contain many self cites (6 of 19) and citations to Crick (5 of 19). At least in the introduction an overview on already published related literature should be given. Eg. The currently published special issue "DNA as Information" http://rsta.royalsocietypublishing.org/content/374/2063. The works by Tlusty:
doi:10.1103/PhysRevLett.100.048101; doi:10.1016/j.plrev.2010.06.002; or - Rumer (see special issue); or - Koonin et al. https://www.ncbi.nlm.nih.gov/pmc/articles/PMC3293468/ and (for all papers) references therein. The here provided list is only a suggestion of possible references and the authors might identify others better fitting to the RNY model.
We thank the reviewer for the literature suggested. Indeed, we added 20 references to provide context to our contribution.
Minor comments
p2, line 42: the abbreviation "tRNA" was used before, please explain abbreviation at the first occurence
In line 52 (introduction) we stated that: “The transfer RNA (tRNA)…”
p5, line 147: The statement "Glycine was the first amino acid" needs a reference.
Eight references (22-29) consider glycine as the first encoded amino acid.
Reviewer 2
This is an interesting manuscript. I suggest to accepting of this manuscript.
Reviewer 3
I am not a mathematician, and therefore a lot of the finer points of this manuscript and its exegesis is admittedly beyond me. However, I am deeply skeptical of the worth of analyzing what is fundamentally a chemical problem – the origin and evolution of the genetic code – using the extremely abstract mathematical approach taken here. In the past such approaches have been of limited utility in providing significant insights into these problems. A famous example of this is the theoretical physicist and cosmologist George Gamow who in the 1950s proposed the extremely elegant Diamond Hypothesis to explain genetic coding, which produced the seemingly miraculous result of getting exactly 20 amino acids from 64 codons. Unfortunately, nature wasn't quite as elegant as he proposed. Interestingly also, at around the same time the physicist-turned-biochemist Francis Crick proposed his 'codons without commas', which was half right, but also half wrong. Which I guess goes to show that biochemists can also be prone to abstract mathematical flights of fancy (or should that be folly?) I must admit I found it hard to grasp – let alone visualise – the relationship between a six-dimensional hypercube and the genetic code. However, I found it troubling that the model reduces the number of possible RNY codes from 2.81 x 10^14 to precisely 1; the phrase too-good-to be-true comes to mind.
It does strike me as interesting that there is an unequal number of amino acids within the four-dimensional hypercube (representing the supposedly ancestral RNY code; Figure 1) which are aminoacylated by Class I and II aaRSs. This is in contrast to the SGC, in which there is a rough 50:50 division between the two classes of aaRSs. A possible explanation for the fact that 75% of the amino acids in the RNY code are aminoacylated by Class II aaRSs is that the latter aminoacylate predominantly small (and hydrophilic) amino acids that likely were the first to be incorporated into an emerging code.
Examples abound in which mathematics and physics have had a great success in biology. Just to mention few examples: the mathematical laws of inheritance by Mendel, the Michaelis-Menten equations of enzyme kinetics, the Hodgkin-Huxley equations for the transmission of nerve impulses, the Neutral theory of molecular evolution by Kimura; The Monod-Wyman-Changeux and the Koshland-Némety-Filmer models for explaining “the second secret of life” about conformational changes that occur during ligand binding to receptors.
The fact that the Diamond code and the comma-less code were incorrect does not mean that our present calculation is incorrect.
I emphasize that our calculation clearly states the assumptions and the result of arriving at only one RNY and SGC, is correct. This is the main value of our contribution: to determine the sufficient and necessary conditions that explain the uniqueness of both the RNY and the SGC.
In lines 119-120 we added the following: “Each dimension describes a type of mutation, transition or transversion acting on each of three bases of any codon. Consequently we obtain the six dimensions.”
Other points:
In line 73 the authors make the bold statement that "the RNY subcode is widely considered as the primeval genetic code", and then gives a single reference in support of this conclusion. While an early RNY code does have reasonably wide support in the field, rather more references would be needed to support its majority position! Again, another explanation would be that the RNY codons in the SGC code predominantly for small, water-soluble amino acids, which were likely early incorporations into the code.
We added references 4-7 that support the RNY subcode as a primeval genetic code. Our work (references 14-16, 30-31, and 35) also support the RNY code.
All in all, we do thank the reviewers for their helpful and acute comments and criticisms; for we feel that they have helped us to improve the quality and presentation of the paper. We expect that the present version of the manuscript answers all their concerns to their satisfaction and the paper can now proceed to its publication.
Yours Sincerely,
Marco V. José PhD
Theoretical Biology Group
Instituto de Investigaciones Biomédicas
Universidad Nacional Autónoma de México
Apartado Postal 70228 Ciudad Universitaria
04510 México D. F., México
Tel/fax= 01-52-555- 622-3894
Author for correspondence: Marco V. José
Email: marcojose@biomedicas.unam.mx; marcovjose57@gmail.com

Reviewer 2 Report
I am not a mathematician, and therefore a lot of the finer points of this manuscript and its exegesis is admittedly beyond me. However, I am deeply skeptical of the worth of analyzing what is fundamentally a chemical problem – the origin and evolution of the genetic code – using the extremely abstract mathematical approach taken here. In the past such approaches have been of limited utility in providing significant insights into these problems. A famous example of this is the theoretical physicist and cosmologist George Gamow who in the 1950s proposed the extremely elegant Diamond Hypothesis to explain genetic coding, which produced the seemingly miraculous result of getting exactly 20 amino acids from 64 codons. Unfortunately, nature wasn't quite as elegant as he proposed. Interestingly also, at around the same time the physicist-turned-biochemist Francis Crick proposed his 'codons without commas', which was half right, but also half wrong. Which I guess goes to show that biochemists can also be prone to abstract mathematical flights of fancy (or should that be folly?) I must admit I found it hard to grasp – let alone visualise – the relationship between a six-dimensional hypercube and the genetic code. However, I found it troubling that the model reduces the number of possible RNY codes from 2.81 x 10^14 to precisely 1; the phrase too-good-to be-true comes to mind.
It does strike me as interesting that there is an unequal number of amino acids within the four-dimensional hypercube (representing the supposedly ancestral RNY code; Figure 1) which are aminoacylated by Class I and II aaRSs. This is in contract to the SGC, in which there is a rough 50:50 division between the two classes of aaRSs. A possible explanation for the fact that 75% of the amino acids in the RNY code are aminoacylated by Class II aaRSs is that the latter aminoacylate predominantly small (and hydrophilic) amino acids that likely were the first to be incorporated into an emerging code.
Other points:
In line 73 the authors make the bold statement that "the RNY subcode is widely considered as the primeval genetic code", and then gives a single reference in support of this conclusion. While an early RNY code does have reasonably wide support in the field, rather more references would be needed to support its majority position! Again, another explanation would be that the RNY codons in the SGC code predominantly for small, water-soluble amino acids, which were likely early incorporations into the code.
Author Response

(The authors gave the same response as above.)

Reviewer 3 Report
This is an interesting manuscript. I suggest to accepting of this manuscript.
Author Response

(The authors gave the same response as above.)

Round 2
Reviewer 1 Report
The authors answered all raised points and from my point of view the manuscript could be accepted for publication in the current form.
Author Response
Thanks for your approval!
Reviewer 2 Report
Unfortunately, I still have major reservations about the usefulness of applying such an abstract mathematical model to the evolution of the genetic code. However, the result obtained by the authors, that the constraints imposed on the system lead to only a single outcome - the modern code - is of wider interest. Could the authors present this result without the hypercube model and associated jargon?Author Response
February 5th, 2017
Ms. Emma Li
Assistant Editor
Manuscript ID: life-170440
Title: On the Uniqueness of the Standard Genetic Code
Special Issue: The Origin and Evolution of the Genetic Code: 100th Anniversary Year of the Birth of Francis Crick
Dear Ms. Emma Li,
Thank you for your email dated February 5th, 2017 and for the referee’s comments.
Please find enclosed with this letter the revised version of the manuscript.
Two reviewers, Reviewer 1 and 2 accept the manuscript. Reviewer 3 suggests discarding the hypercube. Modifications in the manuscript are highlighted in red color, and the answers in this reply letter are marked in red.
Reviewer 3 states the following:
“Unfortunately, I still have major reservations about the usefulness of applying such an abstract mathematical model to the evolution of the genetic code. However, the result obtained by the authors, that the constraints imposed on the system lead to only a single outcome - the modern code - is of wider interest. Could the authors present this result without the hypercube model and associated jargon?”
A scientific paper must be reproducible and therefore que cannot discard the cube model. The articles in which the hypercube has been used are already published. In this work, we make use of a 3-dimensional cube. However, we have added more explanation for those readers who are not familiar with the mathematical terminology or jargon.
We have added the following in section 2. Mathematical model of the RNY code:
“The SGC has been represented in a six-dimensional hypercube [11, 40]. Observing that 64 is equal not only to 43 but also to 26, the codon table can be organized as a 6-dimensional hypercube [30]. In such model the set of codons are treated as the 64 vertices of the hypercube, and they are joint by edges which connect codons that differ by a single nucleotide. Each dimension describes a type of mutation, transition or transversion acting on each of three bases of any codon. Consequently, we obtain the six dimensions.
This symmetrical model [40] can be partitioned exactly into two classes of aaRSs in 6-dimensions; it displays symmetry groups when the polar requirement is used; and the SGC can be broken down into a product of simpler groups reflecting the pattern of degeneracy observed, and the salient fact that evolution did not erase its own evolutionary footsteps. The symmetrical model and the Rodin-Ohno model [13] are one and the same [40].”
In Discussion we added the following:
“The mathematical model plays an important role in the reduction of the possible number of codes. The possible RNY codes were obtained by considering the degeneration in the third base and by assuming that the distribution of aaRRs classes is the same as the current RNY code. Further reductions, up to one code, were only accomplished by the use of our mathematical model.”
We have also added reference [40]:
José, M.V., Zamudio, G. S., Morgado, E. R. A unified model of the standard genetic code. Royal Soc. Open Science. To be published.
All in all, we do thank the three reviewers for their helpful and acute comments and criticisms. We expect that the present version of the manuscript answers all their concerns to their satisfaction and the paper can now proceed to its publication.
Yours Sincerely,
Marco V. José PhD
Theoretical Biology Group
Instituto de Investigaciones Biomédicas
Universidad Nacional Autónoma de México
Apartado Postal 70228 Ciudad Universitaria
04510 México D. F., México
Tel/fax= 01-52-555- 622-3894
Author for correspondence: Marco V. José
Email: marcojose@biomedicas.unam.mx; marcovjose57@gmail.com
